# Unsupervised relation extraction using sentence encoding

Manzoor Ali[1], Muhammad Saleem[2], and Axel-Cyrille Ngonga Ngomo[1]

[1] DICE, Paderborn University,Germany
manzoor@campus.uni-paderborn.de, axel.ngonga@upb.de
[2] AKSW, University of Leipzig, Germany
saleem@informatik.uni-leipzig.de

**Abstract.** Relation extraction between two named entities from unstructured text is an important natural language processing task. In the absence of labelled data, semi-supervised and unsupervised approaches are used to extract relations. We present a novel method that uses sentence encoding for unsupervised relation extraction. We use a pretrained, S-BERT based model for sentence encoding. The model classifies identical patterns using a clustering algorithm. These patterns are used to extract relations between two named entities in a given text. The system calculates a confidence value above a certain threshold to avoid semantic drift. The experimental results show that without any explicit feature selection and independent of the size of the corpus, our system achieves better F score than state-of-the-art unsupervised models.

## 1 Introduction

Relation extraction (RE) is a salient Natural Language Processing (NLP) task, which aims to extract semantic relation between two named entities from natural language text. The relation extraction plays an essential role for many NLP applications such as question answering systems, knowledge bases creation and completion [8] etc. Supervised approaches require labelled data for relation extraction and achieve high performance. However, It is an expansive and tedious task to collect and annotate labelled data[4]. In the absence of training labeled data, un-supervised approaches are used to extract relations from natural language text.

State-of-the-art unsupervised approaches make use of different strategies such as word embeddings [2], entity-type information [9], convolutional neural network [7] etc. to extracts relations from unlabeled corpora. To the best of our knowledge, BERT-based *sentence encoding* is yet not used for relation extraction. It is due to the fact that BERT-based sentence encoding and similarity is computationally highly expensive [5]. We propose a novel unsupervised approach dubbed **US-BERT** that uses the BERT-based sentence encoding [5] on a corpus that is already annotated for named entities(NE). Our main contributions are as follow: We applied BERT-based sentence encoding to extract relations from the unstructured text for the first time to the best of our knowledge; We achieve

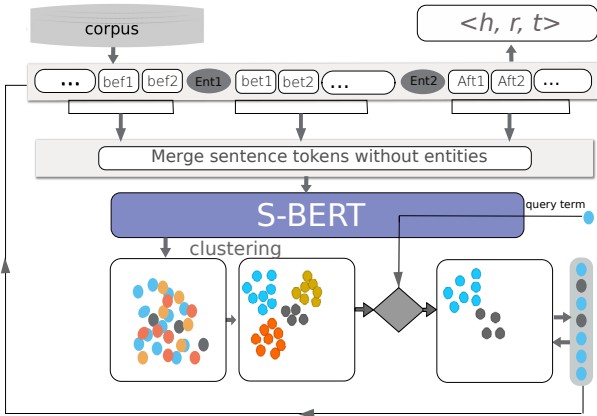

**Fig. 1.** The system Architecture

state-of-the-art results for un-supervised relation extraction; We do not rely on any explicit feature selection for relation extraction.

## 2   Our Approach

Figure 1 describes the US-BERT architecture. Our approach comprises four main modules. The candidate sentences selection module chooses all those sentences that include two already specified entities types. The sentence encoding module uses sentence BERT to calculate 768-dimensional sentence encoding of the selected sentences. The clustering module creates clusters from similar vector representations of sentences. The relation extraction module uses a verb form to choose the most relevant set and extract semantically identical patterns based on cosine similarity.

**Candidate sentences selection:** The proposed model chooses candidate sentences based on NE types for a particular relation to reducing the computation time. A set of candidate sentences $C$ is created from a set of all annotated sentences $S = \{s_1, s_2, s_3...s_n\}$ that include $E_h$ and $E_t$, and the type of $E_h$ and $E_t$ is according to a particular relation. For example for a relation `birthPlace` we only filter those sentences that include entity type Person and Location. $E_h$ and $E_t$ represent the head entity and tail entity, respectively.

The model is based on the context of the sentence. Therefore, we do not consider the entities' direction. Our model uses a window-based approach like Snowball to minimize the false-positive[1]. The window consists of words around two mention entities, `Before` $E_h$, `Between` $E_h$ and $E_t$, and `After` $E_t$. Before sentence encoding, both entities $E_h$ and $E_t$, including entity types, are removed from the sentence.

**Sentence encoding:** We consider the S-BERT based pre-trained model that achieves state-of-the-art performances for sentence encoding [6]. S-BERT uses

siamese and triplet network to produce sentences that are semantically meaningful and also comparable for cosine-similarity. We used pre-trained S-BERT model, *distilbert-base-nli-stsb-mean-tokens*. It is trained on 570,000 sentences.

**Clustering:** Unsupervised algorithms like K-Mean and HAC requires manual selection of the number of clusters. In relation extraction, two entities can have a variable number of relations in the real world. Therefore, we choose Affinity propagation for the selection of centroid and the number of clusters. Affinity propagation does not need the number of cluster variable. It selects an exemplar vector and creates a cluster around the exemplar.

**Query embeddings and relation extraction:** We adopt a query-based approach to extract a relation from a cluster. In our model, a query is a sentence that has two entities in the form of "X and Y". Also, the query contains a relationship in phrasal verb form. For example, the complete query we use for the relation birthPlace is "X born in Y". We use sentence encoding to convert the query to a vector representation. We compute the cosine similarity between all the centroids and the query vector. If the cosine similarity between the centroid of a cluster and query vector crosses the threshold value, we select that cluster for further computation. It helps in the selection of a closely related cluster and also reduces the computation cost.

To increase Recall and avoid semantic drift, we use two iterations. In the first iteration, the system selects only those vectors from a cluster with a high cosine similarity score to the query $P_p$. While in the second iteration, those vectors have high similarity with the list of selected vectors in the first iteration $P_s$. This two-step iteration increases the Recall, but some time causes semantic drift. To avoid semantic drift, we use a threshold value in the first iteration and, in the second iteration, we score the vectors according to the following equation and then compare with the threshold.

$$Pscore = Cosine(P_s) - (1 - Cosine(P_p)) \tag{1}$$

## 3    Evaluation

We evaluate our proposed system on NYT-FB [3] dataset. The NYT-FB dataset is extracted from New York Times articles and aligned with freebase. The NYT-FB dataset consists of 253 relations.

The initial evaluation result of our approach with the state-of-the-art unsupervised systems is shown in Table 1. We run RelLDA1 on their reported parameter on the NYT-FB dataset for only StanfordNER based annotated sentences. In contrast, we run the other two model Simon and EType+ for both StanfordNER and AllenNLP NER. Our proposed model (US-BERT) outperforms all the models in precision for StanfordNER based annotated sentences, but the overall F1 score is less than the EType+. For AllenNLP NER based annotated sentences, our system achieves the highest F1 score. One of the reasons for the low recall we observed is the NER system. Wrongly annotated sentences reduce the recall.

**Table 1.** Precision (P) Recall (R) and F1 score of different systems using two NER annotation techniques on NYT-FB.

| Models | StanfordNER | | | AllenNLP NER | | |
|---|---|---|---|---|---|---|
| | P | R | F1 | P | R | F1 |
| RelLDA1 | 0.30 | 0.47 | 36.8 | - | - | - |
| Simon | 0.32 | 0.50 | 0.39 | 0.334 | 0.497 | 0.399 |
| EType+ | 0.30 | 0.62 | 0.40 | 0.31 | **0.64** | 0.417 |
| **US-BERT** | **0.35** | 0.45 | 0.39 | **0.38** | 0.61 | **0.468** |

## 4 Conclusion and Future Work

We used pre-trained sentence embeddings to extract high-quality relations without any explicit features selection. We achieved the best F1 and precision score when we choose a window-based system. To further investigate the relation extraction, we will use some feature selection, compare the results with our work and see the impact. Overall we achieve state-of-the-art results with our simple and generic approach. In our future work, We will compare our system with some other state-of-the-art systems and datasets.

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
