# OpenReview forum: "Unsupervised relation extraction using sentence encoding"
_eswc-conferences.org/ESWC/2021/Conference/Poster_and_Demo_Track — ESWC2021 P&D_

### Official Review · AnonReviewer1 · 2021-04-13
**Not very novel, but worthy of poster presentation**

**Rating:** 8
**Confidence:** 4

**Review:**

Relation extraction (RE) is a salient Natural Language Processing (NLP) task,
which aims to extract semantic relation between two named entities from natural
language text. The authors propose a novel unsupervised approach dubbed
US-BERT that uses the BERT-based sentence encoding  on a corpus that
is already annotated for named entities. The authors claim to achieve state-of-the-art results.

Overall, I think the work deserves to get presented as a poster, but the claim of the authors that the work is very novel because it applies language models to RE is not a very strong one. LMs have now been used for many difficult problems, include models much newer than BERT. There is nothing exceptional about relation or event extraction. That being said, the directions taken by the work could be useful to the community as a whole.

**Anonymity:**

Yes, I would like my review to remain anonymous.

---

### Official Review · ~Edelweis_Rohrer1 · 2021-04-13
**The paper introduces an unsupervised method that takes advantage of existing named entity annotations in texts, achieving better results in precision and F1 scores that some other approaches.**

**Rating:** 8
**Confidence:** 3

**Review:**

Summary

The paper presents an unsupervised approach named US-BERT, that uses a BERT-based sentence encoding to extract relations between two named entities in a text.
On the one hand, the more common methods for relation extraction are supervised approaches that require annotating the text. On the other hand, unsupervised methods to extract relations from unlabeled texts do not include BERT-based sentence encoding due to it is computationally expensive. The US-BERT approach uses and unsupervised BERT-based sentence encoding but on texts that are already annotated with named entities, without relying on any feature selection. Encouraging results are obtained regarding precision and F1 scores compared with other state-of-the-art systems.

Evaluation

Section 2 presents the system architecture (Figure 1) and describes the four modules which compose the proposed system: the candidate sentence selection module, the sentence encoding module, the clustering module and the relation extraction module.
The diagram describing the architecture in Figure 1 is not clear since it is not possible to distinguish neither the four modules nor interactions among them. However, considering the little space, the responsibilities of each module are clearly described, including some examples.

Final comments

Although the approach would need to be validated by comparing it with other approaches and datasets (future work), the paper introduces an unsupervised method that takes advantage of existing named entity annotations in texts, achieving better results in precision and F1 scores that some other approaches.


**Anonymity:**

No, I would like my review to be deanonymized.

---

### Official Review · AnonReviewer2 · 2021-04-14
**Technical Paper with Poor Exposition**

**Rating:** 5
**Confidence:** 3

**Review:**

The paper presents a BERT-based solution for relation extraction. It assumes that the named entities are given. The approach is interesting and it involves many technicalities (e.g., candidate sentence selection, sentence encoding, clustering, etc.). Unfortunately the paper is not well-written and lacks clarity.

Other comments:
-throughout the paper: “unsupervised” instead of “un-supervised” (“unsupervised” is already used in a proper way a few times)
-page 1: “extract the semantic relation” instead of “extract semantic relation”
-page 1: “However, it” instead of “However, It”
-page 1: “, etc.” instead of “etc.”
-page 1: “named entities (NE)” instead of “named entities(NE)”
-page 2: “entity types” instead of “entities types”
-page 2: “false-positive [1]” instead of “false-positive[1]”
-page 3: explanation of Eq. 1 misses
-page 4: two columns miss the best value in bold

**Anonymity:**

Yes, I would like my review to remain anonymous.

---

### Official Review · ~Julien_Corman2 · 2021-04-14
**The empirical results are promising, but the approach is not clearly motivated.**

**Rating:** 4
**Confidence:** 3

**Review:**

The article describes a system for relation extraction over a corpus with annotated named entities.
The system relies on a BERT model pre-trained for sentences.

The overall approach is relatively simple, and may be summarized as follows.
- The system assumes a set R of predefined binary relations (e.g. "birthplace"), each of which comes with a verbal form (e.g. "X was born in Y" for the relation birthplace).
- The BERT model produces one vector for each of these verbal forms.
- A set S of sentences are selected in the corpus that contain two (exactly or at least, this is unclear) named entities.
- A window of words preceding, in between and following the entities is preserved in each sentence, and the BERT model produces a vector for each of these sequences of words.
- The system compares the vector obtained for each relation r in R to vectors obtained for sentence in S. The more similar the vectors for r and some sentence s in S are (using cosine), the more the relation r is likely to hold between the two named entities identified in s.


The actual approach is a little bit more complex than this summary.
In particular:
- the selection of candidate sentences is based on the types of entities (e.g. "Person" and "Place"),
- sentence vectors are clustered before being compared to a relation vector. The comparison is first performed with the centroids of each cluster, and then with similar vectors within the most relevant clusters.


Interestingly, the approach yields competitive empirical results, even though the BERT model, which is trained for sentence, is not applied to actual sentences, but instead to sequences of words surrounding entities on the one hand, and simple verbal expressions on the other hand.

The empirical results look promising, but unfortunately they are barely discussed.
The description of the system is also vague, and the approach not clearly motivated.

In particular, no justification is provided for the clustering step.
This is arguably confusing: in theory, the vector for a relation may be compared directly to the vectors produced for sentences, without the need to cluster the latter beforehand.
Is clustering a common practice for this form of relation extraction?
If so, some reference would be useful.
If not, some insight would really help to understand this choice.

Similarly, the justification for the specific clustering algorithm that has been used is arguably cryptic ("two entities can have a variable number of relations in the real world. Therefore, we choose Affinity propagation").

The paper also makes repeated usage of the notion of "semantic drift" (e.g. "to avoid semantic drift") to justify some choices, but without further explanation.
As far as I know, "semantic drift" denotes the evolution through time of the meaning of a word/expression.
Do the authors maybe refer to a mismatch between the training corpus of the BERT model and the corpus used for the evaluation?
If so, it is unclear to me how this justifies for instance a threshold value for cosine similarity (Abstract and Page 3).

Overall, I wonder if the format (4 pages) is appropriate to present this type of results.
I feel like a full article in some NLP conference would be a better fit.


## Questions

- Page 2: "choose the most relevant set and extract semantically identical patterns based on cosine similarity."
This is confusing.
What does "set" mean here (maybe clusters)?
And which patterns are extracted (there is no other mention of pattern extraction in the paper)?

- Page 2: "that include two already specified entities types".
Does "two" mean at least or exactly two?
Could the sentence contain more than two named entities, some of which share one of the two types?

- Page 2: "The model is based on the context of the sentence."
What is the context of a sentence?


## Suggestions

- Page 1:
"computationally highly expensive" -> "computationally expensive"

- Page 2: "that include two already specified entities types".
Maybe clarify (even briefly) what "entity type" means here.
From what I understand, the approach assumes that named entities in the corpus are not only detected, but also classified.
If so, it may be useful to state this explicitly (maybe with a reference to a list of type).

- Page 2: "A set of candidate sentences C is created from a set of all annotated sentences S = {s_1 , s_2 , s_3, ..., s_n}".
No need to introduce such a notation: "C", "S" and "s_i" are not used anywhere else in the paper.

- Page 2: "that include E_h and E_t".
Define E_h and E_t beforehand (rather than at the end the paragraph, as it is currently done).

- Page 2: "around two mention entities".
Maybe "around two entity occurrences"?
Or simply "around two entities"?

- Page 2: "head entity and tail entity".
"Head" and "tail" are traditionally used for lists.
Since we are talking about binary relations, "subject and object" or "first and second argument" may be more appropriate.

- Page 3: "a sentence that has two entities in the form of 'X and Y'".
This means that the sentence should contain X, followed by the word 'and', followed by Y.
I think that what is meant here instead is "a sentence that contains two entities X and Y".

- Page 3.
The definition of Pscore would be easier to parse if written:
cos(Ps) + cos(Pp) - 1

- Page 3: "One of the reasons for the low recall we observed is the NER system."
Maybe say explicitly that this did not penalize US-Bert, since all evaluated systems were used with the same NER preprocessing steps.

- Page 4.
I guess precision and recall are computed over the whole corpus, rather than the sentences selected during candidate sentences selection (otherwise the evaluation would be biased).
If so, it would be interesting to quantify how much it impacted recall, i.e. what proportion of false negative are due to this pre-filtering.


## Remarks

- Page 1: "annotate labelled data".
I suspect that what is meant here is "annotate data" (i.e. the data being annotated is not labelled yet), but I am not sure.
The following sentence ("In the absence of training labeled data") suggests that this is the case.
On the other hand, the approach uses corpora where named entities are already identified and typed.

- Page 2: "both entities E_h and E_t, including entity types, are removed from the sentence."
Entity types are annotation, so not part of the sentence.

- Conclusion: "We achieved the best F1 and precision score when we choose a window-based system".
"When we choose" suggests that other (non window-based) options have been investigated.
But no mention is made in the paper.


## Typos

- General.
Inconsistent usage of "labelled" and "labeled".
Choose one of the two, and stick to it.

- General.
"we" and "it", "the" and "recall" should be written in small caps, unless they are the first word of a sentence.

- General:
"un-supervised" -> "unsupervised"

- Abstract:
"F score" -> "F-score"

- Page 1:
"The relation extraction plays an essential role" -> "Relation extraction plays an essential role",
or better, "RE" (introduced in the previous sentence)

- Page 1:
"training labeled data" -> "labeled training data", or simply "training data"

- Page 2:
"to reducing" -> "to reduce"

- Page 3:
"requires manual" -> "require manual"

- Page 3:
"those vectors have high similarity" ->  "those vectors that have high similarity"

- Page 3:
"on NYT-FB [3] dataset" ->  "on the NYT-FB [3] dataset"

- Page 3:
"a set of all annotated sentences that contain" -> "the set of all annotated sentences that contain"

- Page 3:
"the other two model" -> "the other two models"


**Anonymity:**

No, I would like my review to be deanonymized.

---

### Decision · Program_Chairs · 2021-04-19

Accept